# Adult responses to infant prelinguistic vocalizations are associated with infant vocabulary: A home observation study

**Lukas D. Lopez**[1]*, **Eric A. Walle**[1], **Gina M. Pretzer**[1], **Anne S. Warlaumont**[2]

1 Psychological Sciences, University of California, Merced, California, United States of America,
2 Department of Communication, University of California, Los Angeles, California, United States of America

* llopez65@ucmerced.edu

**Data Availability Statement:** The complete dataset and analysis syntax are archived at https://osf.io/a6vps/.

## Abstract

This study used LENA recording devices to capture infants' home language environments and examine how qualitative differences in adult responding to infant vocalizations related to infant vocabulary. Infant-directed speech and infant vocalizations were coded in samples taken from daylong home audio recordings of 13-month-old infants. Infant speech-related vocalizations were identified and coded as either canonical or non-canonical. Infant-directed adult speech was identified and classified into different pragmatic types. Multiple regressions examined the relation between adult responsiveness, imitating, recasting, and expanding and infant canonical and non-canonical vocalizations with caregiver-reported infant receptive and productive vocabulary. An interaction between adult like-sound responding (i.e., the total number of imitations, recasts, and expansions) and infant canonical vocalizations indicated that infants who produced more canonical vocalizations and received more adult like-sound responses had higher productive vocabularies. When sequences were analyzed, infant canonical vocalizations that preceded and followed adult recasts and expansions were positively associated with infant productive vocabulary. These findings provide insights into how infant-adult vocal exchanges are related to early vocabulary development.

## Introduction

Language learning occurs in an inherently social context co-constructed by the infant and their caregivers. Recent research has demonstrated that the quality of infant-caregiver dyadic interactions is important for language learning over and above the quantity of language input [1,2]. For example, while research indicates that the total amount of infant directed speech (IDS) heard in the home is a determinate of language outcomes [3,4], it has also been shown that high-quality in home one-on-one interactions containing IDS are associated with infant vocabulary development over and above the total amount of IDS heard [5,6]. Such dyadic contexts may propagate contingent infant-caregiver vocal turn-taking containing infant speech-

**Funding:** ASW contribution was supported by the National Science Foundation (BCS-1529127; SMA-1539129) and James S. McDonnell Foundation Scholar Award in Understanding Human Cognition.

**Competing interests:** The authors have declared that no competing interests exist.

related vocalizations, which have been shown to drive language outcomes in laboratory observations [7,8]. However, how such infant speech-related vocalizations and contingent IDS in the home promotes language outcomes remains understudied.

In the lab, contingent infant-caregiver turn-taking interactions have been associated with a host of infant language outcomes. Specifically, contingent and sensitive adult responses to infant vocal initiations are associated with more advanced infant vocal and linguistic development in later months [9]. One possible explanation for these effects is that contingent adult responses, such as those occurring during turn taking episodes, lead infants to match the phonological features of the adult vocal responses or learn the words for objects in the environment as a function of the type of caregiver response [7,8,10–12]. Indeed, specific pragmatic types (e.g., imitation, naming, imperatives) of contingent parental feedback to infant speech-related vocalizations have been shown to promote interactions associated with receptive and productive vocabulary development [7,8,13]. These findings suggest that specific contingent IDS responses to infant speech-related vocalizations foster infant word learning and promote production of speech-related sounds [14,15].

While laboratory research examining the importance of parent contingent responding to infant speech-related vocalizations is prevalent, more naturalistic observations of such patterns is necessary to assess whether the findings generalize across contexts. Specifically, the naturalistic home environment is a necessary context in which to observe such vocal interactions because it captures more of the variation in infant-caregiver communication over different contexts, activities, and routines inherent to daily life and the natural learning ecology of the infant [16]. The present investigation hand-coded samples taken from day-long home audio recordings to examine how speech-related infant vocalizations, contingent adult IDS of varying pragmatic functions, and contingent patterns of their cooccurrence were associated with infant vocabulary size.

## Infant vocal development

The quality of infants' vocalizations progresses markedly during the first year of life. From birth, infants produce reflexive vocalizations, such as cries, fusses, and vegetative noises (e.g., coughs, burps, sneezes), as well as primitive prespeech (a.k.a. "protophone") vocalizations [17–19]. By three months of age, infants typically demonstrate an expanded range of vocal types, such as raspberries, squeals, growls, full vowels, yells, whispers, and primitive consonant-vowel syllables known as marginal babbling [17,18]. At around 7 months infants typically begin to demonstrate canonical babbling, which contains both consonants and vowels with swift transitions between them. Canonical babbling is considered speech-like and is a foundation for the first words that infants begin producing around the first birthday [18,20].

The communicative value of pre-linguistic infant vocalizations should not be downplayed. For example, the presence of an engaging adult affects the quantity and quality of infant vocalizations [7,21], demonstrating the social nature of these behaviors. Infants' prelinguistic vocalizations also demonstrate functional flexibility, i.e., the ability to produce different types of vocalizations with different affective meanings across contexts, which is an essential feature of language [22]. Furthermore, infant speech-related vocalizations have been shown to play an important role in evoking caregiver verbal responses that are crucial for infant vocabulary development [14,23]. These findings accentuate the importance of considering the communicative nature of infant vocalizations when examining subsequent caregiver responsiveness in bidirectional language learning contexts.

## Contingent adult responses

On the other side of the conversational coin is how parents respond to and initiate infant vocalizations. Caregiver contingent responses to infant speech-like vocalizations facilitate infant language development [7] and caregiver verbal initiation of turn taking with infants as young as 3 months of age results in more speech-like sounding infant vocalizations [10,24]. This patterning of dyadic vocal interaction has been portrayed as a social feedback loop wherein adults are more likely to reply contingently to infants' speech-like vocalizations and infants are more likely to produce speech-like vocalizations following contingent adult speech [15,23,25]. Analyzing two- and three-event vocal sequences has utility in measuring the reciprocity of these vocal exchanges [26,27]. Nonetheless, adult contingent responses can take a variety of forms and serve different functions.

Parsing out how pragmatically distinct adult responses are related to infant language development can provide further nuance to the study of dyadic vocal interactions. One behavior that adults use in responding to young infants' prelinguistic utterances is vocal matching, or imitation. For example, mothers of young infants have been observed to match the complexity of their infants' precanonical vocalizations [12]. Additionally, caregivers vary in the types of responses that they provide across infancy. Tamis-LeMonda, Bornstein, and Baumwell [28] observed that parents primarily used affirmations and imitations with young infants, and then expanded their responding to include expansions, descriptions, and questions as their children's abilities increased. Clearly, caregivers are sensitive to infant vocal capacities and respond accordingly, thereby fostering infant vocal development.

Furthermore, sensitive maternal responses following infant vocalizations and gestures early in development are associated with infants' later maternal-directed vocalizations and progression through specific language milestones [9,13]. Adults selectively reinforce different types of infant vocalizations, responding with more play vocalizations to infant vowel sounds, whereas they more often imitate in response to infant consonant-vowel vocalizations [14]. Moreover, certain types of caregiver responses to infant speech-like vocalizations have been shown to facilitate language learning. Specifically, contingent responses containing object labels to infant consonant-vowel vocalizations help infants learn word-object associations [8,13] and contingent caregiver responses with consonant-vowel utterances promote infant matching of their caregiver's vocal complexity [7]. Thus, specific caregiver responses to quality infant vocalizations are intricately linked with infant language outcomes.

## Contexts for studying infant-adult contingent responding

To date, the majority of studies examining the relation of specific types of caregiver responses with certain kinds of infant vocalizations have been restricted to laboratory settings under a limited range of conditions [7,9,10,13,14,23]. Although lab sessions can be similar to in-home interactions, qualitative and quantitative differences across these observational contexts exist [16,29,30]. Moreover, studies that do occur in the home environment often rely on a researcher being present [28,31,32], which results in a less ecologically valid environment [33].

Researchers are increasingly utilizing audio recordings collected over the course of a full day while children are at home and no researchers are present, thereby minimizing observer effects [33]. Sampling the language environment in this way also allows for a broader array of contexts to be analyzed, thus yields a more accurate view of infants' typical experiences. Further, assessing infant learning in such contexts provides a window into how infant-caregiver interactions naturally progress over time and the behavioral ecologies in which they usually occur, including diverse cultural contexts and caregiving practices that are understudied in

developmental psychology but consistently emphasized in fields such as cultural psychology and anthropology [34,35]. However, a major obstacle of such long-form home recordings is the labor intensiveness of coding such lengthy segments. Thus, previous research using such recordings has understandably relied on the use of automated coding [25]. Unfortunately, in doing so it is not possible to analyze various infant vocalization types, parent response types, and more general social behaviors, which necessitate meticulous coding by a human listener. Furthermore, the examination of the temporal contingencies between specific infant vocalizations types and IDS necessitates the use of human coding [27]. The result of the above challenges is that while we know that one-on-one infant-caregiver interactions featuring IDS in the home fosters vocabulary development [5,6], the specific roles of distinct infant vocalization types and caregiver responses, and their contingency upon one another in these interactions, has yet to be examined in naturalistic home observations. It is this chasm between ecologically valid language environment sampling and meticulous coding of contingent infant-caregiver interactions that is bridged in the present investigation.

## The current study

This study utilized LENA digital language processors to capture daylong audio recordings of 13-month-old infants' home language environments, from which a total of six distinct 5-mintue segments (30 minutes total) of audio were hand-coded for each infant. This strategy allowed us to capture unique episodes of the infant's natural language ecology, increasing the range of activities and settings sampled. Infant speech-related vocalizations were coded as canonical or non-canonical, and adult vocalizations directed to the infant were coded for their pragmatic functions (naming, description, question, directive, prohibition, imitation, recast, and/or expansion; codes based on [14]). The temporal unfolding of infant and adult vocalizations of each type were then analyzed to examine how specific sequences of infant-adult and infant-adult-infant vocalization sequences related with infant vocabulary sizes.

We hypothesized that pragmatically distinct forms of IDS would relate to infant receptive and productive vocabulary over and above the total amount of IDS. Specifically, we predicted that IDS that provided labels of objects in the environment (i.e., naming) would be positively associated with infant receptive vocabulary. Additionally, we hypothesized that IDS that contained an adult like-sound response to the preceding infant vocalization (i.e., imitations, recasts, expansions) would be positively associated with infant productive vocabulary. We also examined how the above variables interacted with the quality of infant vocalizations. We predicted that increased infant canonical babbling occurring in conjunction with caregivers demonstrating more of the IDS pragmatic codes of interest would be positively associated with infant vocabulary size.

We also analyzed specific infant-adult-infant turn-taking episodes. We hypothesized that interactions featuring infant canonical vocalizations with the contingent adult responses of interest would be positively associated with infant vocabulary. Based on previous laboratory research [7], we also hypothesized that adult like-sound responses initiated and responded to by infant canonical vocalizations would be associated with infant productive vocabulary. Specifically, we hypothesized that three-event sequences following the pattern of "infant speech-like vocalization—adult like-sound response—infant speech-like vocalization" would be associated with infant productive vocabulary.

The coding protocol, and additional results and figures are available in the Supplemental Online Material (SOM). This study is registered (https://osf.io/a6vps/), and the SOM, data, syntax, and coding protocol are on OSF.

## Method

### Participants

Fifty-three infants (24 female; Mage = 12.80 months, SD = 0.59, range = 11.64–14.50) and their mothers completed the study. All 19 infants from a previous study [27] were included in this sample. All families were recruited from the California San Joaquin Valley and had primarily English-speaking households (i.e., caregivers reported that English was spoken to the infant at least 50% of the day). Parents' average level of education was a college degree (no high school degree (n = 2), high school degree (n = 20), college diploma (n = 22), graduate degree (n = 9)), and mothers self-identified as Caucasian (n = 26), Hispanic (n = 21), Asian (n = 4) and Black (n = 2). Infants had no prior diagnosis of developmental disorders or hearing impairment.

An additional 50 families from the larger study were excluded prior to coding because their recording was not returned or shorter than the 10-hour requirement (n = 24), the primary language spoken in the home was Spanish (i.e., > 50% of the time, according to the parents' estimates; n = 17), or the recording was split over multiple days (n = 9). The requirement to record 10 hours within a single day was designed to help ensure that the samples were drawn from a range of contexts that the infant experienced over the course of a day.

### Procedure

All procedures were approved by the University of California Merced Institutional Review Board: UCM13-0006. Written consent was obtained from the parents of all participants.

Each family received the study materials by postal mail or personal delivery to their home. The materials included a LENA recording device, a LENA vest to hold the recorder, and a packet of questionnaires, including a demographic form and the MacArthur-Bates Developmental Inventory: Words and Gestures [36]. Adults were instructed to put the vest on the infant, place the recorder inside, and record a typical day (e.g., no parties or special trips) of the child's language environment. The LENA device recorded up to 16 hours of audio and captured all infant vocalizations, nearby adult vocalizations, and other nearby environmental noises. Adults had the option to exclude portions that contained personal information.

### Coding

**Sample selection.**   The audio recordings were first processed using the LENA Pro software (LENA; LENA Research Foundation, Boulder, Colorado, United States). The software's automatic labeling system applied various sound source labels to the recording, including vocalizations (target infant, male adult, female adult, other child, and overlap) and non-human sounds (e.g., television or radio, noise, silence) and has demonstrated reasonable reliably locating target adult vocalizations (82% agreement with human coders) and target infant vocalizations (78% agreement with human coders) [37,38]. It also identifies "conversational turns", i.e. instances where an infant and an adult vocalization were separated by 5 s or less of silence, noise, electronic sounds, overlap, or another child. LENA's Automatic Data Extractor (ADEX) relies on these automatically generated sound source labels to determine the number of adult vocalizations, target infant vocalizations, conversational turns, and other event types that occur in specified time windows and provides an Excel export of the corresponding counts in each time segment. ADEX was used to identify the 3 most voluble infant (i.e., containing the highest number of infant vocalizations) and 3 most voluble interactive (i.e. containing the highest number of conversational turns between the infant and caregiver) 5-minute samples from each infant's recording. Additionally, the following criteria were implemented in the

selection of the samples: (1) the sample began no less than 15 minutes before the onset or after the offset of all other included samples to ensure that unique episodes were sampled across the daylong audio recording (average of 1 sample replaced per infant); (2) a human listener confirmed that the sample included at least 10 infant-speech related vocalizations (i.e., canonical or non-canonical utterances; 6 samples replaced overall); and (3) the infant was judged to not have an object obstructing their mouth (e.g., a pacifier) for the majority of the sample (1 sample replaced overall). In the event a criterion was not met, the next most voluble infant or interactive sample, respectively, was chosen and subjected to the same criteria. This procedure continued until 3 high infant volubility and 3 high conversational turn count samples were identified for each infant and ensured that each sample was representative of observational times across the entire day, the automatically assessed high volubility of the sample was valid, and the infant's vocal expressivity was not impeded. The majority of the selected samples occurred during infant play or meal times spread across the entire day for all participants.

**Hand-coding.** A total of 30 minutes (six 5-minute samples) was hand-coded for each participant. For each 5-minute sample, infant and adult vocalizations were marked by five primary coders using ELAN software [39]. Reliability coding of infant and adult vocalization types for a random 30% of the samples was performed by the first author who was naïve to the primary codes. Reliability was performed by categorizing already located adult and infant vocalizations into their corresponding types.

**Locating vocalizations.** Researchers marked vocalizations by listening to the recording and pausing the clip when a vocalization was heard. Vocalizations were identified as produced by either the target infant, an adult, or another child. Older siblings were present in approximately 40% of the segments. When confusion arose in regard to identifying target infant vocalizations, coders met with the first author to listen to the segment together and identify the target infant. The onset and offset of each vocalization was then marked. Vocalizations were defined as "any sound of nontrivial loudness that you think was made by the vocal tract." Coders were instructed to code boundaries so that "annotations run from the onset of sound to the offset of sound".

Training involved each of the five primary coders being given a 30-minute sample containing 132 infant vocalizations from the beginning of a daylong sample of a 13-month-old infant in the current study that was coded previously by the first and third author. Training consisted of independently coding 5 minutes from the sample, and then comparing one's own codes to a "gold standard" set of consensus codes from that sample coded by the first and third authors. Each coder met with the first author to receive feedback each time the codes were compared. This was repeated three times before the coder could begin coding on their own. The dates in which coders began and finished training are available in the supplemental materials, following the document containing the written instructions that coders were given. Only infant and caregiver vocalizations were coded further (see Table 1 for specific definitions and examples of infant and caregiver codes and the Cohen's kappa reliability metrics for each group of codes).

**Infant vocalization type codes.** Infant vocalizations were coded as *Reflexive* (laugh and cry), *Vegetative* (e.g., burp, hiccup, cough, yawn, heavy breathing, etc.), *Non-Canonical/Non-Reflexive* (including marginal babbling and other non-canonical protophones; henceforth referred to as "Non-Canonical"), or Canonical (i.e., containing at least one syllable that has speech-like timing between consonant and vowel; [17]). Coder training included reading the phonetic catalog of infant pre-speech vocalizations [17], coding 15 minutes from the gold standard sample, and discussion with the authors. Reflexive and vegetative infant vocalizations were not considered further. Reliability for the infant vocalization codes demonstrated substantial inter-rater agreement ($\kappa$ = .63; [40]) consistent with previous coding of this nature [41] and a good percent reliability (percent agreement = 86.58%) comparable to previous

Table 1. Infant and adult vocalization type codes.

| Infant Vocalization (κ = .63) | Definition | |
|---|---|---|
| Reflexive | Laugh or cry | |
| Vegetative | Burp, hiccup, cough, yawn, sneeze, breathing | |
| Non-Canonical | Marginal babbling and other non-canonical protophones, including fuss | |
| Canonical | A vocalization containing at least one syllable that has both a true consonant and a full vowel with speech-like timing of the CV or VC transition | |
| Caregiver Environmental (κ = .67) | Definition | Example |
| Naming | Providing a label for an object | "That's a ball" |
| Description | Explaining features or functions of an object | "That's a big red ball" |
| Question | Asking the infant a question | "Do you want down?" |
| Directive | Telling the infant to do something | "Put on your shirt" |
| Prohibition | Inhibiting the infant from acting | "Don't touch that" |
| Other | Any other kind of language | "That's a good job" |
| Caregiver Like-Sound (κ = .82) | Definition | Example |
| Imitation | Mimicking an infant speech-related vocalization | Infant says, "ba" and the adult responds, "ba." |
| Recast | Responding to the infant speech-related vocalization with a like sounding real word | Infant says, "ba" and the adult responds, "ball." |
| Expansion | Responding to the infant's speech-related vocalization with a like sounding real word and providing more information or context | Infant says, "ba" and the adult responds, "yes, that is a big red ball." |

literature [7]. Although this type of coding is admittedly difficult to code with high reliability, it is worth the undertaking given the interest in these variables for language development [7], and hand coding remains the gold standard for annotating infant vocalization types containing canonical syllables [42].

**Caregiver utterance type codes.** Vocalizations made by the caregiver were first coded as either *infant-directed*, *other-directed* (i.e., speaking to another person who was not the target infant), or *unknown*. Research assistants determined caregiver direction by inferring who the intended recipient of the adult vocalization was. All coders heard examples of IDS compared to ODS in the gold standard training to demonstrate the nature of IDS, such as an elongated higher pitch pattern typical of IDS or whether the semantic content addressed the infant (e.g., saying their name). Although utterance direction is more difficult to code using audio compared to video recordings, it has still been reliably accomplished by human coders [43,44]. Caregiver direction codes demonstrated substantial inter-rater agreement (percent agreement = 90.18%; κ = .80; [38]).

Next, the function of each IDS vocalization by the caregiver was categorized based on the verbal content of the utterance using codes inspired by descriptions from [14]. Function codes were separated into two categories, environmental or adult like-sound (see Table 1 for definitions and examples). The environmental function codes were: (1) naming, (2) description, (3) question, (4) directive, (5) prohibition, and (6) other. The adult like-sound function codes were: (1) imitation, (2) recast, and (3) expansion. The codes were mutually exclusive within categories, but could overlap between categories (e.g., an adult vocalization of "ball" after infant vocalization of "ba" could be coded as naming [environmental code] and as a recast

[adult like-sound code]). The IDS function codes demonstrated substantial inter-rater agreement for environmental codes (percent agreement = 81.20%; κ = .67) and near perfect agreement for adult like-sound codes (percent agreement = 91.40%; κ = .82; [40]).

For a given 5-minute segment, the same coder listened to the entire segment and coded both adult utterance direction and IDS function in the same coding pass.

**Sequence codes.**   Finally, three-event sequence codes (i.e., infant vocalization—adult utterance—infant vocalization) centered around each caregiver utterance were created to identify sequences where specific infant vocalization types were followed by specific adult response types that were subsequently followed by specific infant vocalization types. The three-event sequences analyzed were chosen based on sequences hypothesized to be relevant based on prior research [10,14,25,26]. These analyses served to complement analyses focusing separately on the frequency of specific response types and the frequency of specific infant vocalization types by relating children's vocabulary levels to *sequences* of specific infant and adult behaviors. Rather than attempt to detect causal effects of infant vocalizations on adult responses and vice versa, the goal was to relate frequency of occurrence of specific sequences to infant vocabulary size.

The function codes and their corresponding timestamps were exported into a separate data file. Research assistants located each caregiver IDS utterance and identified whether an infant vocalization occurred before and after the utterance. Non-overlapping infant vocalizations occurring within two seconds before or after a caregiver infant-directed speech utterance were categorized as infant preceding and infant response, respectively. Instances of IDS utterances without a preceding or following speech-related infant vocalization were bounded by "blank" codes. This process resulted in 3-event sequences that consisted of infant initiations followed by caregiver infant-directed speech codes and ending in infant responses (e.g., canonical—recast—canonical). Lastly, two-event sequence codes were derived from the three-event codes by adding all of the instances starting with a particular infant vocalization type followed by a particular adult vocalization type (e.g., canonical—adult like-sound responding two event sequences were generated from the 3-event sequences of: canonical—adult like-sound response—non-canonical, canonical—adult like-sound response—canonical, and canonical—adult like-sound response—blank) to determine if contingent infant—caregiver interactions associated with infant vocabulary.

## Infant vocabulary size

Infant vocabulary was assessed using the MacArthur-Bates Communicative Development Inventory: Level 1 (MCDI) [36]. Parents were instructed to fill out all 396 items in the MCDI for words that their infant "understands" (i.e., receptive vocabulary) and "understands and says" (i.e. productive vocabulary). Although parent reports of infant vocabulary naturally contain some degree of reporting bias [45], analyses of the internal validity (.95) and test-retest reliability (.87) for the MCDI are strong [46], and studies have shown that the language assessment yields comparable patterns as in lab evaluations [47,48]. Two caregivers did not fill out the receptive vocabulary portion of the MCDI and were thus excluded from analyses on infant receptive vocabulary but included in productive vocabulary analyses.

## Analytic strategy

**Preliminary analyses.**   The six samples (3 highest infant volubility and 3 highest turn count samples) for each infant were collapsed in the analyses. First, we report the mean rates of infant vocalizations, caregiver responsiveness, and infant MCDI scores for comparison with prior research using these measures. Next, we report bivariate correlations that were

conducted to assess whether the hypothesized variables of interest were associated with infant vocabulary. Specifically, we assessed if adult naming and infant canonical vocalizations were associated with infant receptive vocabulary, and if adult like-sound responding IDS codes (imitation, recast, expansion) and infant canonical vocalizations were associated with productive vocabulary. Adult like-sound response IDS codes were initially collapsed and analyzed as a single variable to be consistent with previous literature, which has grouped these responses (i.e., imitation, recast, expansion) as a single variable [9,14,28]. We then analyzed the three distinct types of adult like-sound responses (i.e., imitation, recast, expansion) to determine how each was associated with productive vocabulary. A Bonferroni correction was applied to the correlational analyses to identify variables of interest that were correlated with vocabulary (adjusted, p = .004). All variables correlated at $.004 < \alpha < .05$ were tested using the same regression models as those used to test the variables of interest. None of these models were significant (see supplemental materials).

**Event frequency analyses.** Hierarchical multiple regressions tested the unique associations of the variables of interest with infant vocabulary size. All models included control variables in Step 1, specifically infant age due to its significant correlation with infant vocabulary, total IDS to control for the total amount of caregiver input, and total infant vocalizations to control for infants who were more voluble. The results remain consistent without control variables and when controlling for socioeconomic status and caregiver response rates (see SOM). For models examining infant receptive vocabulary size, Step 2 included the frequency of adult naming, frequency of infant canonical vocalizations, and their interaction predicting receptive vocabulary. Models testing associations with infant productive vocabulary size were carried out in two phases. The first phase used the collapsed adult like-sound responding variable, frequency of infant canonical vocalizations, and their interaction term to predict productive vocabulary size. The second phase separately examined how each type of adult like-sound response code (i.e., imitation, recast, expansion) was associated with productive vocabulary.

**Event sequences analyses.** Lastly, we examined whether contingent temporal sequences of infant vocalizations and IDS codes of interest were associated with infant productive vocabulary. Step 1 of each model included the control variables (i.e., infant age, total IDS, total infant vocalizations). Step 2 included base rates of the variables of interest to ensure that the results were specific to the sequences and not an artefact of the base rates of the components of the sequences. Specifically, because infant canonical vocalizations and adult like-sound responding demonstrated significant positive associations with productive vocabulary, the base rates of these variables were included in Step 2 of the infant canonical—adult like-sound sequence models predicting productive vocabulary to control for infant overall speech-like vocalization production and parent overall responsiveness.

As mentioned in the introduction, analyzing two- and three-event vocal sequences are a way to assess reciprocal vocal exchanges [26,27]. We initially used two-event sequences to determine whether infant canonical or non-canonical vocalizations preceding instances of adult like-sound responding were positively associated with infant word production. Next, based off these findings, the aforementioned interesting communicative role of infant canonical vocalizations, and previous research supporting a feedback loop for scaffolding infant word production [25], we examined canonical three-event sequences to determine if canonical vocalizations occurring before, after, or before and after instances of adult like-sound responding predicted productive vocabulary. Again, two phases of analyses were conducted, the first examining sequences involving the collapsed adult like-sound responding variable, and the second examining sequences with each subtype of adult response (i.e., imitation, recast, expansion).

## Results

### Preliminary analyses

Infants produced an average of 39.11 (SD = 17.15) speech related vocalizations per 5-minute segment, consisting on average of 10.44 (SD = 8.44) canonical vocalizations and 28.67 (SD = 14.79) non-canonical vocalizations. The proportion of babbling that was canonical (.27) compared to that which was noncanonical (.73) babbling was comparable to canonical babbling ratios (CBR) reported at the syllable level in previous lab research with same aged infants (CBR = .27 to .30; [49]; CBR = .34; [41]), and, not surprisingly, higher than canonical babbling ratios reported for 11-month old infants in similar naturalistic observations (CBR = .11; [29]). Additionally, caregivers responded to 21 percent of infant speech-related vocalizations on average, notably lower than rates reported in lab observations [14,50]. Lastly, parent report of infant receptive (M = 104.59, SD = 71.65, range: 3–293 words) and productive (M = 13.21, SD = 18.79, range: 2–93 words) vocabulary size on the MCDI were consistent with previously reported findings with this age group [48,51].

Zero-order correlations used a Bonferroni correction to $\alpha$ = .004 to adjust for the number of comparisons. It was revealed that infant receptive vocabulary was only significantly associated with infant age, $r(51)$ = .39, $p$ = .004, and total IDS, $r(51)$ = .43, $p$ = .002. The association of receptive vocabulary with caregiver naming, $r(51)$ = .35, $p$ = .01, and infant canonical vocalizations, $r(51)$ = .15, $p$ = .29, did not reach statistical significance. Bonferroni corrected zero-order correlations indicated that infant productive vocabulary was significantly associated with infant canonical vocalizations, $r(53)$ = .41, $p$ = .002, and adult like-sound responding, $r(53)$ = .45, $p < .001$ (specifically recasts, $r(53)$ = .47, $p < .001$, and expansions, $r(53)$ = .48, $p < .001$, but not imitation, $r(53)$ = .21, $p$ = .13). No other significant associations were observed among coded variables with receptive and productive vocabulary scores ($p$'s > .005) (see Table 2).

### Receptive vocabulary

**Frequency of speech type.** Hierarchical multiple regression was used to examine the relation of adult naming, infant canonical vocalizations, and the Adult Naming x Infant Canonical

**Table 2. Correlations of coded variables with receptive and productive vocabulary.**

|  | Receptive Vocabulary | Productive Vocabulary |
|---|---|---|
| Infant Age | .39** | .38* |
| Infant Canonical Vocalizations | .15 | .41** |
| Infant Non-canonical Vocalizations | .04 | -.04 |
| IDS Total | .43** | .33* |
| Adult Naming | .35* | .16 |
| Adult Description | .25 | .27 |
| Adult Question | .29* | .19 |
| Adult Directive | .20 | .22 |
| Adult Prohibition | .00 | .17 |
| Adult Like-Sound Responding | .32* | .45** |
| Imitation | .26 | .21 |
| Recast | .24 | .47** |
| Expansion | .23 | .48** |

Note:

** = $p < .004$,

* = $p < .05$.

Vocalizations interaction term with infant receptive vocabulary size. Step 1 revealed significant main effects of infant age, $b = 41.38$, $p = .01$, and the total amount of IDS, $b = 1.90$, $p = .004$, but not total infant vocalizations, $b = -0.19$, $p = .72$. Step 2 showed no significant effects of adult naming, $b = 0.77$, $p = .90$, infant canonical vocalizations, $b = -1.36$, $p = .36$, nor the Adult Naming x Canonical Vocalizations interaction, $b = 0.25$, $p = .66$. Because none of the variables of interest were significantly associated with infant receptive vocabulary, no further analyses were pursued. However, we did include the sequence models for receptive vocabulary in the supplemental materials and they were not significant.

### Productive vocabulary

**Frequency of speech type.** Infant productive vocabulary size was analyzed in two stages: first using the collapsed adult like-sound responding variable, and second examining subtypes of adult like-sound responses (i.e., imitation, recast, expansion). Hierarchical multiple regression was used to assess the relation of adult like-sound responding, infant canonical vocalizations, and the Adult Like-Sound Responding x Infant Canonical Vocalizations interaction term with infant productive vocabulary size (see Table 3). Significant main effects of infant age, $b = 10.72$, $p = .01$, and total IDS, $b = 0.38$, $p = .03$, were present in Step 1, but there was no significant effect of total infant vocalizations, $b = 0.08$, $p = .95$. Step 2 indicated no significant main effects of infant canonical vocalizations, $b = 0.09$, $p = .79$, nor adult like-sound responding, $b = 1.63$, $p = .37$, but there was a significant Adult Like-Sound Responding x Infant Canonical Vocalizations interaction, b = 0.42, p < .001, 95% CI [0.19, 0.64] (see Table 3, Step 2a). As shown in Fig 1, infants demonstrating high levels of canonical vocalizations and receiving high adult like-sound responding had larger productive vocabularies ($p = .004$).

**Type of adult like-sound response and infant canonical vocalizations.** To further tease apart the adult like-sound responses variable, we examined the relations of each adult like-sound response subtype (i.e., imitation, recast, and expansion) to determine how each was associated with productive vocabulary (see Table 3 for integrated results).

**Imitation.** Hierarchical multiple regression examined the relation of imitation, infant canonical vocalizations, and the Imitation x Infant Canonical Vocalizations interaction term with infant productive vocabulary size (see Table 3, Step 2b). The main effect of imitation was not significant, $b = 1.09$, $p = .68$. However, significant effects were present for infant canonical vocalizations, $b = 0.90$, $p = .01$, and the Imitation x Infant Canonical Vocalizations interaction, $b = 0.80$, $p = .02$, 95% CI [0.11, 1.49].

**Recast.** Hierarchical multiple regression of recasts, infant canonical vocalizations, and the Recast x Infant Canonical Vocalizations interaction term predicting infant productive vocabulary size revealed no significant main effects of adult recasts, $b = 1.83$, $p = .72$, nor infant canonical vocalizations, $b = 0.02$, $p =. 97$. However, the Recast x Infant Canonical Vocalizations interaction was significant, $b = 0.52$, $p = .01$, 95% CI [0.12, 0.93] (see Table 3, Step 2c).

**Expansion.** Hierarchical multiple regression examined the relation of expansion, infant canonical vocalizations, and the Expansion x Infant Canonical Vocalizations interaction term with infant productive vocabulary size (see Table 3, Step 2d). Results indicated no significant main effects of adult expansions, $b = -3.23$, $p = .66$, nor infant canonical vocalizations, $b = 0.10$, $p = .75$, but a significant Expansion x Infant Canonical Vocalizations interaction was present, $b = 1.57$, $p < .001$, 95% CI [0.77, 2.37].

**Event sequences analyses.** Finally, we assessed how two and three event sequences involving the variables of interest related to infant productive vocabulary. Step 1 included the control variables as described in the Analytic Strategy section. First, the two-event sequences of infant non-canonical—adult like-sound response and of infant canonical—adult like-sound response

**Table 3. Multiple regression with adult like-sound responding predicting productive vocabulary.**

| Control Variables | Productive Vocabulary | |
|---|---|---|
| | $\beta$ | $\Delta R^2$ |
| Step 1 | | .23* |
| Infant Age | .34* | |
| Total IDS | .29* | |
| Total Infant Vocalizations | .08 | |
| Step 2a | | .26** |
| Adult Like-Sound Responding | .17 | |
| Infant Canonical Vocalizations | .05 | |
| Adult Like-Sound Responding X Infant Canonical Vocalizations | .54** | |
| Type of Adult Like-Sound Response | | |
| Step 2b | | .14** |
| Imitation | .08 | |
| Infant Canonical Vocalizations | .41** | |
| Imitation X Infant Canonical Vocalizations | .29* | |
| Step 2c | | .17** |
| Recast | .08 | |
| Infant Canonical Vocalizations | .01 | |
| Recast X Infant Canonical Vocalizations | .44** | |
| Step 2d | | .26** |
| Expansion | -.06 | |
| Infant Canonical Vocalizations | .05 | |
| Expansion X Infant Canonical Vocalizations | .60** | |

Note:

\*\* = $p < .01$,

\* = $p < .05$.

were used to predict productive vocabulary. Again, significant main effects of infant age, $b = 10.72$, $p = .01$, and total IDS, $b = 0.38$, $p = .03$, were present in Step 1, with no significant main effect of total infant vocalizations, $b = 0.01$, $p = .95$. However, of greater interest, Step 2 identified a positive main effect of adult like-sound responding, $b = 12.11$ $p = .02$, 95% CI [2.03, 22.19], yet a significant negative association of infant non-canonical vocalization—adult like-sound responding pairings, $b = -3.41$, $p = .007$, 95% CI [-5.84, -0.98]. No significant associations were observed in these two-event sequence analyses for infant canonical vocalizations, $b = -0.15$, $p = .75$, nor canonical—adult like-sound responses pairings, $b = -0.11$, $p = .92$. Thus, infants who more frequently heard an adult like-sound response following their non-canonical vocalization had smaller productive vocabularies.

Next, each of the four possible combinations of infant canonical and adult like-sound response three-event sequences (i.e., non-canonical—adult like-sound response—canonical; canonical—adult like-sound response—blank; canonical—adult like-sound response—non-canonical; and canonical—adult like-sound response—canonical) were analyzed to determine their specific relations with infant productive vocabulary (see Table 4). Step 1 in each of these models was identical to the previous models of productive vocabulary. Step 2 indicated that infant productive vocabulary size was significantly associated with only the canonical—adult like-sound response—canonical sequence, $b = 2.49$, $p = .04$, 95% CI [0.09, 4.90]. None of the other three-event sequences were significant (all $p$'s > .24) and thus they were not examined further.

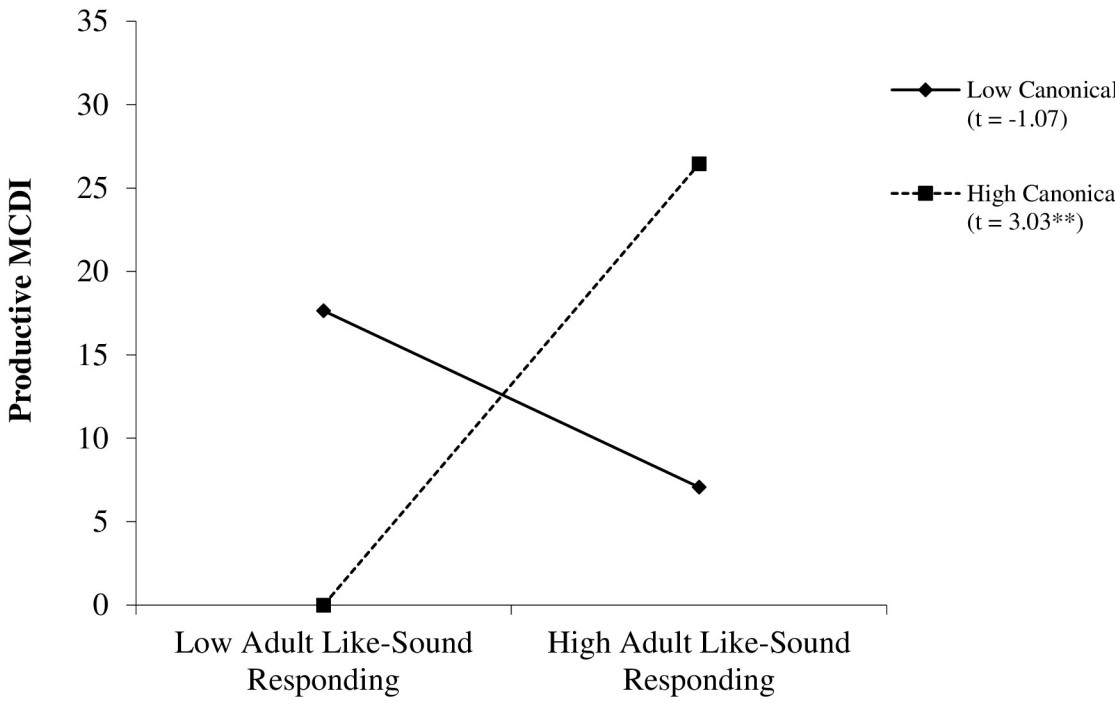

**Fig 1. Adult like-sound responding and infant canonical vocalizations predicting infant productive vocabulary.** Infant productive vocabulary for low (-1 *SD* from the mean) and high (+1 *SD* from the mean) levels of Adult Like-Sound Responding (low = 0.20, high = 3.55 per 5-minutes) and Infant Canonical Vocalizations (low = 2.00, high = 18.88 per 5-minutes) based on regression analysis presented in Table 3. Numbers in parentheses are unstandardized simple slopes. (** = $p \leq$ .01).

**Type of adult like-sound response canonical sequences.** Finally, to parse out whether a specific type of adult like-sound response accounted for the above effect, we examined how the frequency of specific adult like-sound response types (i.e., imitation, recast, expansion) bounded by infant canonical vocalizations predicted infant productive vocabulary size (see Table 4 and Fig 2 for integrated results).

**Imitation.** Hierarchical multiple regression examined the relation of imitation, infant canonical vocalizations, and the canonical—imitation—canonical sequence with infant productive vocabulary size (see Table 4, Step 2b). There were no significant effects of adult imitation, $b = 0.02$, $p = 1.00$, infant canonical vocalizations, $b = 0.63$, $p = .12$, nor the canonical—imitation—canonical sequence, $b = 1.92$, $p = .54$.

**Recast.** Hierarchical multiple regression of recasts, infant canonical vocalizations, and canonical—recast—canonical sequence predicting infant productive vocabulary size revealed no significant main effects of adult recasts, $b = -0.55$, $p = .92$, nor infant canonical vocalizations, $b = 0.06$, $p =. 89$. However, the canonical—recast—canonical sequence was significant, $b = 4.09$, $p = .03$, 95% CI [0.51, 7.67] (see Table 4, Step 2c).

**Expansion.** Hierarchical multiple regression examined the relation of expansion, infant canonical vocalizations, and the canonical—expansion—canonical sequence with infant productive vocabulary size (see Table 4, Step 2d). Results indicated no main effects of adult expansions, $b = 4.68$, $p = .54$, nor infant canonical vocalizations, $b = 0.13$, $p = .75$, but the canonical—expansion—canonical sequence was significant, $b = 7.08$, $p = .04$, 95% CI [0.37, 13.80].

**Table 4. Multiple regression with adult like-sound sequences predicting productive vocabulary.**

| Control Variables | Productive Vocabulary | |
|---|---|---|
| | $\beta$ | $\Delta R^2$ |
| Step 1 | | .23* |
| Infant Age | .34* | |
| Total IDS | .29* | |
| Total Infant Vocalizations | .08 | |
| Step 2a | | .22** |
| Adult Like-Sound Responding | .32 | |
| Infant Canonical Vocalizations | .04 | |
| Non-Canonical—Adult Like-Sound Response—Canonical | -.03 | |
| Canonical—Adult Like-Sound Response—Blank | -.16 | |
| Canonical—Adult Like-Sound Response—Non-Canonical | -.24 | |
| Canonical—Adult Like-Sound Response—Canonical | .55* | |
| Type of Adult Like-Sound Response | | |
| Step 2b | | .08 |
| Imitation | .00 | |
| Infant Canonical Vocalizations | .28 | |
| Canonical—Imitation—Canonical | .16 | |
| Step 2c | | .16** |
| Recast | -.02 | |
| Infant Canonical Vocalizations | -.03 | |
| Canonical—Recast—Canonical | .56* | |
| Step 2d | | .15** |
| Expansion | .11 | |
| Infant Canonical Vocalizations | .06 | |
| Canonical—Expansion—Canonical | .35** | |

Note:

** = $p < .01$,

* = $p < .05$.

## Discussion

This study utilized naturalistic observations of infant-caregiver interactions to explore how the quantity and quality of infant vocalizations and caregiver responses were related with concurrent caregiver-reported infant vocabulary size. Below we review the present findings with regards to how different types of IDS, infant vocalizations, and sequences of infant-caregiver and infant-caregiver-infant vocal interactions related with infant language.

### The role of pragmatics in the IDS-vocabulary relationship

Specific types of infant-directed speech were associated with infant productive vocabulary size, over and above the total amount of IDS the infant heard. Specifically, infant productive vocabulary size was positively associated with adult responses to infant speech-related vocalizations that incorporated sounds having been produced by the infant. Our results indicated that adult use of imitations, recasts, and expansions were each positively associated with infant productive vocabulary size. These findings mirror previous laboratory research [9] and expand research emphasizing the role of caregiver vocal matching with younger infants [7,11]. Further, our analyses of vocal sequences highlight the functionally distinct roles that imitations,

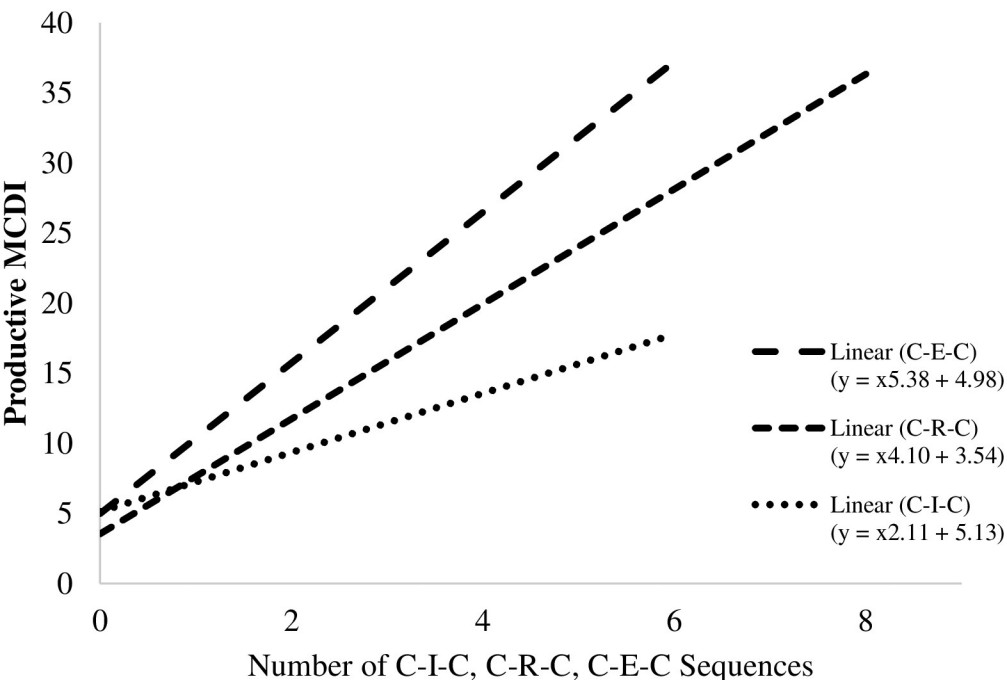

**Fig 2. Adult like-sound canonical sequences association with infant productive vocabulary.** Linear associations between the total raw frequencies of Canonical—Imitation—Canonical (C-I-C), Canonical—Recast—Canonical (C-R-C), and Canonical—Expansion—Canonical (C-E-C) sequences and Infant Productive MCDI scores. See Supplemental Online Materials for scatterplots.

recasts, and expansions play in facilitating language development. Interestingly, findings indicate that adult use of recasts and expansions in response to infant canonical vocalizations positively associated with productive vocabulary. These findings differentiate between response types typically grouped together [14,28]. For example, Nicholas, Lightbown, and Spada [52] describe recasts as serving to correct language and provide more information for school age children. Our results provide some additional preliminary support for this notion, providing evidence that recasts and expansions play such a role in the early stages of word production and may facilitate future productive language outcomes in infancy.

## Adult responses to infant speech-like vocalizations

The role of infant vocalization type preceding the adult like-sound response was also found to be related with infant language. Specifically, infants who heard more adult like-sound responding and who also produced more canonical vocalizations had larger productive vocabularies. This may be because infant speech-like vocalizations were easier for adults to recast and expand upon. Research with older children has found that adults are more likely to correct a well-formed utterance than one of lesser quality and impaired children who have a difficulty forming well-formed sentences receive less recasting [53,54]. Thus, adults may find it easier to respond meaningfully to infant vocalizations that are closer to intended words, thereby strengthening infants' associations between their own vocalizations and potentially meaningful communicative functions of those vocalizations. This interpretation is supported by our finding that infants who produced more canonical vocalizations and had caregivers providing frequent adult like-sound responses had larger productive vocabularies (see Fig 1). Indeed, prior

research suggests that adult scaffolding of higher-quality infant vocalizations creates a positive feedback loop, whereas reinforcement of lower quality infant vocalizations may actually hinder infant speech production [15]. Additionally, infant canonical vocalizations toward the caregiver may establish a kind of vocal play routine. Infants produce higher quality vocalizations towards caregivers in order to initiate play routines and show a willingness to learn or practice new vocalizations [8,9]. The vocal motor learning facilitated by such play may help develop the motor speech skills necessary to produce recognizable words. It is also possible that caregivers who respond with recasts and expansions are more likely to attend to infant speech-like vocalizations and thus could have reported more infant speech productions as meaningful words on the caregiver-reported vocabulary measure.

## Infant-adult vocalization sequences

Our analyses of two- and three-sequence codes of infant-caregiver interactions provided a more nuanced look at the functions of caregiver feedback to infant canonical vocalizations. Infant non-canonical vocalizations preceding an adult like-sound response had a negative association with productive vocabulary, supporting the above notion that providing contingent feedback to lower quality infant vocalizations may actually be detrimental to infant vocabulary development. Interestingly, however, infant canonical, but not non-canonical, vocalizations occurring before and after the adult like-sound response were associated with productive vocabulary. This finding adds to a growing body of literature indicating that infants' speech-like vocalizations provide greater communicative value and influence caregiver behavior over other prelinguistic vocalizations [55,56].

Previous research has noted the link between infant vocal development and how the caregiver responds [9,13,28]. Our findings relating infants' vocal behaviors and specific types of adult responses in naturalistic settings corroborate and extend previous laboratory research linking infant-adult vocal interaction with infant vocabulary. Specifically, adult recasts and expansions preceded by and responded to with infant speech-like vocalizations were positively associated with productive vocabulary. The conversational nature of these interactions supports a theoretical social feedback loop wherein high-quality infant vocalizations elicit high-quality adult responses, which in turn elicit high-quality infant responding [15,23,25]. Interestingly, our findings demonstrate that the social feedback loop may have differential functions for adult imitation of infant vocalizations compared to recasting and expanding upon them (see Fig 2 for an illustration). Adult recasts and expansions bounded by infant canonical vocalizations may reflect a routine in which the infant practices producing a word and the adult provides contingent feedback with correction as appropriate, thereby facilitating subsequent higher quality infant vocalizations and productive language [57]. This may be of particular importance when considering the dyadic mechanisms that contribute to language development in both typical and atypical populations. For example, evidence suggests that children who have difficulty producing certain speech sounds may benefit from corrective feedback provided by adult recasts so that they may better hear the difference in pronunciation [58]. Additionally, emphasis on contingent caregiver responses may have social benefits for children struggling with social interactive skills, such as children with autism who have been found in at least one study to receive less contingent caregiver responses than their typically developing peers on average [25 though see 59]. Ultimately, analysis of distinct sequential patterns of infant vocalizations and adult IDS can provide key information about the co-construction of the infant-caregiver home language environment.

## Limitations and future directions

Although the present study replicated laboratory findings regarding infant-caregiver vocal interactions and extended such research to a naturalistic home environment, some limitations warrant discussion. First, contrary to our hypothesis, caregiver naming and infant canonical vocalizations were not significantly associated with infant receptive vocabulary. Further analysis of the context in which caregiver naming instances occur could shed light on this null finding. Specifically, although the majority of the samples in the current study took place during play and mealtimes, more nuanced analyses of the types of routines dyads were engaged in during each segment would be insightful. For example, maternal responsiveness and speech types have been shown to vary over different contexts, routines, play sessions, and toy sets [50,60,61]. Additionally, coding infant vocalizations from home video recordings would further our understanding of contextual factors involved in such vocal interactions. Previous research has shown that when infants produce an object-directed canonical babble and receive a subsequent caregiver naming instance, the infant is more likely to learn the object name [8].

Second, although our coding and measures were based on prior research and demonstrated substantial reliability, some methodological concerns may exist. For instance, the parent-report nature of the MCDI could introduce some reporting bias and the possible association between parent verbal responsiveness and reporting of child language cannot be entirely ruled out. However, recent work has demonstrated that the MCDI is a reliable assessment of vocabulary size [62] and can be a valid indicator of language delays as evidenced through other measures [63]. It is conceivable that our data reflect a feedback loop wherein parents begin interpreting their child's productions as verbal, which encourages them to interact with expansions and recasts to their infant's canonical vocalization, which supports the infant's further productive vocabulary development. This could make it impossible to identify a single causal pathway explaining the present study's results. Yet even if this scenario holds, it would also mean that the present results reflect real causal associations between infant-adult interaction patterns and productive vocabulary development. Nonetheless, future work should incorporate non-parent-report methodologies to assess whether observable measures of infant vocabulary replicate the present findings.

Finally, the dynamic nature of development makes it essential to assess the associations of dyadic interactions with infant vocabulary across time. While the present study suggests that infant-adult interactions and infant vocabulary are linked around the infant's first birthday, these behaviors undoubtably have downstream consequences for language development [13,28]. For example, assessing infant vocabulary size at later ages would clarify the long-term impact of the observed interactions. Further, this study only scratches the surface of examining the temporal dynamics of infant-caregiver dyadic interactions. Utilizing a methodology that naturally accounts for base rates of vocalizations, such as computing a Reciprocal Vocal Contingency score, would help determine if vocal sequences are related above chance occurrences [26]. Moreover, it is likely that the sequences of conversational turns analyzed in the present study extended beyond the binary and trinary sequence codes that were imposed. Assessing naturalistic interactions of longer turn taking sequences, as well as how such patterns of interaction may vary across hours or days, would further our understanding of how such bidirectional patterns are associated with infant language development [27]. Methodological techniques, such as Recurrence Quantification Analysis, could be utilized toward this end [64]. We encourage future research exploring the temporal dynamics and broader contextual factors of dyadic interactions so that we may obtain a fuller understanding of infant language development.

## Acknowledgments

This work was supported by the National Science Foundation (BCS-1529127; SMA-1539129) and James S. McDonnell Foundation Scholar Award in Understanding Human Cognition. We gratefully thank the research assistants in the Interpersonal Development Lab and the Emergence of Communication Lab as well as the infant participants and their families.

## Author Contributions

**Conceptualization:** Lukas D. Lopez.

**Formal analysis:** Lukas D. Lopez.

**Writing – original draft:** Lukas D. Lopez.

**Writing – review & editing:** Eric A. Walle, Gina M. Pretzer, Anne S. Warlaumont.

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
