## [Decision Letter · Decision Letter 0]

24 Aug 2020

PONE-D-20-19792

Adult responses to infant prelinguistic vocalizations are associated with infant vocabulary: A home observation study

PLOS ONE

Dear Dr. Lopez,

Thank you for submitting your manuscript to PLOS ONE. I have now received two reviews and although both reviewers agree that the manuscript has merit they have raised a few issues that I feel should be addressed. I have decided for major revision since one of the reviewers has proposed additional analysis. 

Therefore, we invite you to submit a revised version of the manuscript that addresses the points raised during the review process, which can be summarised as follows:

1) Consider a stronger motivation of the study highlighting the importance of ecologically valid observations and the overlooked role of context in language learning.

2) Reviewer 2 raises a methodological question about the use of sequential analysis. Please either consider further analysis or reply justifying your decision. Reviewer 2 also points to a couple of data that would interesting to report such as the number of vocalisations occurring which were not responded to by the mother.

3) Both reviewers suggest some additions to the discussion, such as linking to existing literature considering the role of activity context and also considering potential applications of your research. 

Please go over the reviews and reply to all of the comments and questions raised, either in the manuscript or in your rebuttal letter.

We look forward to receiving your revised manuscript.

Kind regards,

Iris Nomikou, Ph.D.

Academic Editor

PLOS ONE

Journal Requirements:

2. Please provide additional details regarding participant consent.

In the ethics statement in the Methods and online submission information, please ensure that you have specified whether consent was informed.

Reviewers' comments:

Reviewer's Responses to Questions

**Comments to the Author**

1. Is the manuscript technically sound, and do the data support the conclusions?

Reviewer #1: Yes

Reviewer #2: Yes

2. Has the statistical analysis been performed appropriately and rigorously? 

Reviewer #1: Yes

Reviewer #2: Yes

3. Have the authors made all data underlying the findings in their manuscript fully available?

Reviewer #1: Yes

Reviewer #2: Yes

4. Is the manuscript presented in an intelligible fashion and written in standard English?

Reviewer #1: Yes

Reviewer #2: Yes

5. Review Comments to the Author

Reviewer #1: Thank you for the opportunity to review the manuscript titled “Adult response to infant prelinguistic vocalizations are associated with infant vocabulary: A home observation study”

The study focused on relations between adult responses to infant vocalizations in the context of a naturalistic home environment using LENA. I think the study is very valuable and adds to prior literature on parent-infant interactions using ecologically valid methods. A few questions and suggestions came up for me while reading the manuscript that I think, if addressed, could strengthen the paper.

First, I think the framing and motivation of the paper isn’t strong enough and could use more nuance. The study is motivated by the naturalistic method, suggestions that such method is more ecologically valid than the structured lab environment. I wholeheartedly agree. There aren’t enough studies focused on what parents actually do in the confines of their homes and how that shifts across the day. However, it isn’t that lab studies don’t generalize at all or capture something completely artificial and different from naturalistic environments. Lab studies, rather, capture a specific aspect of interactions that is found in the home environment. Tamis-LeMonda and colleagues’ paper (citation 16 in the manuscript) on methods shows that structured observations (which were done in the parents home, but are similar to structured observations in lab environments) capture the highest consecutive five minutes of interaction in the home. They capture the chattiest moments, that is. It is no surprise that then years of studies on language input during structured observations predict child language development. it isn’t the the lab methods are invalid, they do capture something meaningful. They just don’t capture ecology, fluctuation of talk across the day, the setting and activity within which interactions take place and learning occurs, the most interesting and embodied aspects of learning. The aspects of children’s context that are largely ignored by developmental psychologists who focus on language but are constantly articulated by cultural psychologists and linguistic anthropologists are necessary. This point I think is important to address in the introduction, and necessitates a stronger motivation for the study to be articulated by the authors.

Second, information presented about contingent responsiveness is incomplete. It seems that the authors captured base rates of responses, that is, frequencies of how many times parents responded to the different types of vocalizations. In work on contingent responsiveness, typically researchers would go a step further and conduct sequential analyses to gather probabilities of mother responding to infant vocalizations. Such probabilities account for base rates of behaviors, and can be used in analyses predicting language outcomes. These probabilities matter because they differentiate mothers who respond to an infant twice (but infants’ total vocalizations are 10) from a mother who responds to an infant twice (but infants’ total vocalizations are 3). The second mother is more responsive than the first. On line 305 the authors state that they are not seeking to detect causal effects of infants’ vocalizations, which is a slight mischaracterization of the baseman’s GSEQ method. They are probabilities, and they take into account all of the mother and infant talk. On page 29, the authors use the language “elicit” which appears causal anyhow. So my questions are: if authors are trying to extend prior literature into the home and see if patterns found in structured observations in lab (or home!) mirror, why not use established methods for contingent responsiveness? Regardless, the authors should then also report how many infant vocalizations occurred in each segment that the mothers did not respond to to give readers a sense of the overall interaction. In the context of the home, unlike the structured context, mothers may attend to other tasks, and interactions may be fluid, comprised of smaller bouts perhaps. I would presume that mothers would have lower rates of responding to total number of infant vocalizations in the home than in the lab, where they have little else to do but interact, and my hypothesis would also be that bouts of very contingent interactions would focus on specific activities, like book sharing or play.

Finally, I appreciate that the authors noted the types of activities that occurred in the segments chosen, and I think that those should be highlighted. The study focuses on language interactions in the context of everyday life, and so interpretation of input in the context of play specifically (which comprised most segments it appears) is meaningful and necessary. It should also then be linked to prior work on language input by activity context, perhaps in the discussion.

Reviewer #2: This is a well written manuscript that details the findings for in home parent-infant vocal interactions and their relation to language outcomes. This is an area of research that is underrepresented in the literature, as the majority of studies have been conducted in the lab. The meticulous coding captures specific responses (recasts, expansions) to more advanced infant vocalizations (canonical vocalizations) that support language development. The findings reveal important moment to moment vocal exchanges that can support positive outcomes. I appreciate the limitations discussed by the authors, but I would also suggest that there are potential applications of their research that they could elude to in the discussion. I agree that these are just short snippets of exchanges, but the importance of uncovering which specific vocalizations and feedback (loops) is significant for understanding how parents support language in day to day interactions.

6. PLOS authors have the option to publish the peer review history of their article (what does this mean?). If published, this will include your full peer review and any attached files.

Reviewer #1: **Yes: **Yana Kuchirko

Reviewer #2: No

---

## [Author Response · Author response to Decision Letter 0]

16 Sep 2020

9/15/2020

Dear Editor Nomikou,

We submit for your consideration a revision of our manuscript, titled “Adult responses to infant prelinguistic vocalizations are associated with infant vocabulary: A home observation study,” submitted as a Research Article to PLOS ONE. In what follows, our responses to each point are detailed in italics, with the reviewers’ comments given in quotes.

Thank you once again for your thoughtful consideration of our work, and for the opportunity to improve it. We feel that the paper has been improved with the benefit of the feedback that we have received and look forward to your assessment. 

Sincerely,

Lukas Lopez (on behalf of all co-authors)

Department of Psychological Sciences

University of California, Merced

Merced, CA 95343, USA

Email: llopez65@ucmerced.edu

 

Replies to Reviewer: 

Reviewer #1: 

“First, I think the framing and motivation of the paper isn’t strong enough and could use more nuance. The study is motivated by the naturalistic method, suggestions that such method is more ecologically valid than the structured lab environment. I wholeheartedly agree. There aren’t enough studies focused on what parents actually do in the confines of their homes and how that shifts across the day. However, it isn’t that lab studies don’t generalize at all or capture something completely artificial and different from naturalistic environments. Lab studies, rather, capture a specific aspect of interactions that is found in the home environment. Tamis-LeMonda and colleagues’ paper (citation 16 in the manuscript) on methods shows that structured observations (which were done in the parents home, but are similar to structured observations in lab environments) capture the highest consecutive five minutes of interaction in the home. They capture the chattiest moments, that is. It is no surprise that then years of studies on language input during structured observations predict child language development. it isn’t the the lab methods are invalid, they do capture something meaningful. They just don’t capture ecology, fluctuation of talk across the day, the setting and activity within which interactions take place and learning occurs, the most interesting and embodied aspects of learning. The aspects of children’s context that are largely ignored by developmental psychologists who focus on language but are constantly articulated by cultural psychologists and linguistic anthropologists are necessary. This point I think is important to address in the introduction, and necessitates a stronger motivation for the study to be articulated by the authors.”

We thank the reviewer for the positive assessment of the methodologies used in the present paper and the excellent suggestion to further highlight the behavioral ecology of the observed behaviors. We agree that while in-lab observations capture meaningful caregiver behaviors, studies in the naturalistic home environment capture the ecology in which these behaviors normally transpire, including a much greater range of settings and activities such as cultural contexts and caregiving practices. We now reference work from cultural psychology and anthropology that highlight these points and have emphasized the utility of naturalistic recordings for capturing a variety of learning ecologies in the Introduction of the revised manuscript (p. 4, lines 66-69; p. 7, lines 137-142; & p. 8, lines 160-163).

“Second, information presented about contingent responsiveness is incomplete. It seems that the authors captured base rates of responses, that is, frequencies of how many times parents responded to the different types of vocalizations. In work on contingent responsiveness, typically researchers would go a step further and conduct sequential analyses to gather probabilities of mother responding to infant vocalizations. Such probabilities account for base rates of behaviors, and can be used in analyses predicting language outcomes. These probabilities matter because they differentiate mothers who respond to an infant twice (but infants’ total vocalizations are 10) from a mother who responds to an infant twice (but infants’ total vocalizations are 3). The second mother is more responsive than the first. On line 305 the authors state that they are not seeking to detect causal effects of infants’ vocalizations, which is a slight mischaracterization of the baseman’s GSEQ method. They are probabilities, and they take into account all of the mother and infant talk. On page 29, the authors use the language “elicit” which appears causal anyhow. So my questions are: if authors are trying to extend prior literature into the home and see if patterns found in structured observations in lab (or home!) mirror, why not use established methods for contingent responsiveness?” 

We thank the reviewer for this suggestion and for pointing out the places where we have inadvertently somewhat misrepresented the prior literature and been sloppy in our use of terminology implying causal relationships!

We agree that matching previous methodologies will indeed increase the generalizability about how behaviors observed in the lab translate to the naturalistic home environment. Past studies relating parent-infant interactions in more constrained settings to infant vocabulary and language outcomes have used a variety of different methodologies. Sometimes the focus has been on controlling for base rates and other times it has not, and when it has controlled for base rates, the methods used have varied. The models in the present paper controlled for the overall number of infant vocalizations and caregiver infant-directed utterances in Step 1 and for the overall number of infant canonical vocalizations and caregiver responses in Step 2, thus controlling for the base rates of infants’ vocalizations and caregivers’ responsiveness. We have added additional text in the revised manuscript to clarify how we controlled for the base rates of these vocal behaviors (p. 19, lines 409-410). 

We have edited the text in the Sequence Codes subsection of the Method section so that it no longer implies that sequential analyses (such as GSEQ) intend to detect causal relationships. We have also replaced the first instance of “elicit” in the Discussion section from the prior submission with a statement that implies less causality. We did, however, keep the term when discussing the potential feedback loop that we believe the results support—but we added mention that that feedback loop is “theoretical” (p. 30, lines 630-635).

“Regardless, the authors should then also report how many infant vocalizations occurred in each segment that the mothers did not respond to to give readers a sense of the overall interaction. In the context of the home, unlike the structured context, mothers may attend to other tasks, and interactions may be fluid, comprised of smaller bouts perhaps. I would presume that mothers would have lower rates of responding to total number of infant vocalizations in the home than in the lab, where they have little else to do but interact, and my hypothesis would also be that bouts of very contingent interactions would focus on specific activities, like book sharing or play.”

We agree with the reviewer that this would be useful descriptive information to present in the paper. We have now included the average rate in which caregivers contingently responded to infant speech-related vocalizations in the observed segments (p. 20, lines 435-436).

Additionally, controlling for caregiver response rates in Step 1 in all of the models does not change the results. We have added this information in text (p. 18, lines 392) and have added the output for the models controlling for caregiver response rates to the SOM (see S12 & S13). 

Furthermore, we have added the number of infant vocalizations that did and did not receive contingent caregiver responses for each infant to the updated dataset on OSF. Additionally, we added a ZIP file that contains all of the timestamped excel exports from which these data were derived. Specifically, each export contains timestamped infant vocalization and caregiver response codes for each participant. This way researchers interested in probing these contingencies further have all of the data at their disposal to do so. 

“Finally, I appreciate that the authors noted the types of activities that occurred in the segments chosen, and I think that those should be highlighted. The study focuses on language interactions in the context of everyday life, and so interpretation of input in the context of play specifically (which comprised most segments it appears) is meaningful and necessary. It should also then be linked to prior work on language input by activity context, perhaps in the discussion.”

The reviewer raises an excellent point that the contexts in which the interactions took place may have important implications of our findings. We have added further discussion of this point in the Future Directions section of the Discussion (p. 31, lines 666-670). 

Reviewer #2: 

“This is a well written manuscript that details the findings for in home parent-infant vocal interactions and their relation to language outcomes. This is an area of research that is underrepresented in the literature, as the majority of studies have been conducted in the lab. The meticulous coding captures specific responses (recasts, expansions) to more advanced infant vocalizations (canonical vocalizations) that support language development. The findings reveal important moment to moment vocal exchanges that can support positive outcomes. I appreciate the limitations discussed by the authors, but I would also suggest that there are potential applications of their research that they could elude to in the discussion. I agree that these are just short snippets of exchanges, but the importance of uncovering which specific vocalizations and feedback (loops) is significant for understanding how parents support language in day to day interactions.”

We thank the reviewer for noting the importance of the current findings. The revised manuscript now includes additional potential applications of how the present results may inform caregiver practices that facilitate positive language outcomes in the home environment in the Discussion (p. 28, lines 586-588 & ps. 30-31, lines 642-656).

---

## [Decision Letter · Decision Letter 1]

29 Oct 2020

Adult responses to infant prelinguistic vocalizations are associated with infant vocabulary: A home observation study

PONE-D-20-19792R1

Dear Dr. Lopez,

The reviewers and myself have now viewed the revision of the manuscript. We’re pleased to inform you that your manuscript has been judged scientifically suitable for publication and will be formally accepted for publication once it meets all outstanding technical requirements.

Kind regards,

Iris Nomikou, Ph.D.

Academic Editor

PLOS ONE

Additional Editor Comments (optional):

Reviewers' comments:

Reviewer's Responses to Questions

**Comments to the Author**

1. If the authors have adequately addressed your comments raised in a previous round of review and you feel that this manuscript is now acceptable for publication, you may indicate that here to bypass the “Comments to the Author” section, enter your conflict of interest statement in the “Confidential to Editor” section, and submit your "Accept" recommendation.

Reviewer #1: All comments have been addressed

2. Is the manuscript technically sound, and do the data support the conclusions?

Reviewer #1: Yes

3. Has the statistical analysis been performed appropriately and rigorously? 

Reviewer #1: Yes

4. Have the authors made all data underlying the findings in their manuscript fully available?

Reviewer #1: Yes

5. Is the manuscript presented in an intelligible fashion and written in standard English?

Reviewer #1: Yes

6. Review Comments to the Author

Reviewer #1: It was a pleasure reading the revision of this manuscript. All of my questions have been addressed. I think the manuscript will make a valuable contribution to the literature on contingent responsiveness.

7. PLOS authors have the option to publish the peer review history of their article (what does this mean?). If published, this will include your full peer review and any attached files.

Reviewer #1: **Yes: **Yana Kuchirko

---

## [Editor Report · Acceptance letter]

3 Nov 2020

PONE-D-20-19792R1 

Adult responses to infant prelinguistic vocalizations are associated with infant vocabulary: A home observation study 

Dear Dr. Lopez:

I'm pleased to inform you that your manuscript has been deemed suitable for publication in PLOS ONE. Congratulations! Your manuscript is now with our production department. 

Kind regards, 

on behalf of

Dr. Iris Nomikou 

Academic Editor

PLOS ONE